# Historical Mitogenomic Diversity and Population Structuring of Southern Hemisphere Fin Whales

**DOI:** 10.3390/genes14051038

**Published:** 2023-05-03

**Authors:** Danielle L. Buss, Lane M. Atmore, Maria H. Zicos, William P. Goodall-Copestake, Selina Brace, Frederick I. Archer, C. Scott Baker, Ian Barnes, Emma L. Carroll, Tom Hart, Andrew C. Kitchener, Richard Sabin, Angela L. Sremba, Caroline R. Weir, Jennifer A. Jackson

**Affiliations:** 1British Antarctic Survey, National Environment Research Council, Cambridge CB3 0ET, UK; 2Department of Archaeology, University of Cambridge, Downing Street, Cambridge CB2 3DZ, UK; 3Centre for Ecological and Evolutionary Synthesis, Department of Biosciences, University of Oslo, 0316 Oslo, Norway; 4The Natural History Museum, Cromwell Road, London SW7 5BD, UK; 5Scottish Association for Marine Science, Oban PA37 1QA, UK; 6National Oceanic and Atmospheric Administration, Southwest Fisheries Science Center, La Jolla, CA 92037, USA; 7Marine Mammal Institute and Department of Fisheries and Wildlife, Hatfield Marine Science Center, Oregon State University, Newport, OR 97365, USA; 8Te Kura Mātauranga Koiora—School of Biological Sciences, University of Auckland Waipapa Taumata Rau, Auckland 1010, New Zealand; 9Department of Zoology, University of Oxford, Mansfield Road, Oxford OX1 3SZ, UK; 10Department of Natural Sciences, National Museums Scotland, Chambers Street, Edinburgh EH1 1JF, UK; 11School of Geosciences, University of Edinburgh, Drummond Street, Edinburgh EH8 9XP, UK; 12Falklands Conservation, Ross Road, Stanley F1QQ 1ZZ, Falkland Islands

**Keywords:** baleen whale, population structure, genomic analysis, South Pacific, South Atlantic, ancient DNA

## Abstract

Fin whales *Balaenoptera physalus* were hunted unsustainably across the globe in the 19th and 20th centuries, leading to vast reductions in population size. Whaling catch records indicate the importance of the Southern Ocean for this species; approximately 730,000 fin whales were harvested during the 20th century in the Southern Hemisphere (SH) alone, 94% of which were at high latitudes. Genetic samples from contemporary whales can provide a window to past population size changes, but the challenges of sampling in remote Antarctic waters limit the availability of data. Here, we take advantage of historical samples in the form of bones and baleen available from ex-whaling stations and museums to assess the pre-whaling diversity of this once abundant species. We sequenced 27 historical mitogenomes and 50 historical mitochondrial control region sequences of fin whales to gain insight into the population structure and genetic diversity of Southern Hemisphere fin whales (SHFWs) before and after the whaling. Our data, both independently and when combined with mitogenomes from the literature, suggest SHFWs are highly diverse and may represent a single panmictic population that is genetically differentiated from Northern Hemisphere populations. These are the first historic mitogenomes available for SHFWs, providing a unique time series of genetic data for this species.

## 1. Introduction

Baleen whales are important top predators in marine ecosystems that provide an array of ecosystem services. This includes exerting top-down effects on marine food webs during predation [1] and providing vital organic and often limiting nutrients (e.g., iron) to many species through defecation and at the end of life by providing vital nutrients to many deep-sea, often endemic, species as whale falls [1,2,3]. Fin whales, *Balaenoptera physalus*, are the second largest species of baleen whale in the world. In the Southern Hemisphere (SH), fin whales have a broad circumpolar and subpolar foraging distribution [4] and play a vital role in Southern Ocean ecosystems [5,6,7]. They undertake seasonal migrations between low and high latitudes [8] and make longitudinal movements within and between ocean basins [9,10], providing ecosystem services across a broad range of habitats and likely interacting with a multitude of different species [1].

During the industrial whaling era—spanning from 1902 to 1986 in the SH—fin whales were the most heavily exploited species (in terms of absolute abundance), with over 700,000 fin whales killed during the 20th century [11,12]. While it has been suggested that the sudden removal of baleen whales from the marine environment during the 20th century had lasting ecological repercussions [2], the ecosystem impacts of fin whale removal are still poorly understood. Recent research has highlighted the importance of understanding the genomic population structure in assessing biodiversity loss and population declines as well as in informing conservation management [13]. Thus, in order to understand the impact of whaling on Southern Hemisphere fin whale (SHFW) populations and provide conservation and management advice as fin whales recover from whaling, a proper understanding of the genomic population structure is crucial.

Three subspecies of fin whale are currently recognised: *B. physalus physalus* in the North Atlantic, *B. physalus velifera* in the North Pacific, and *B. physalus quoyi* in the SH [14]. A putative additional subspecies, *B. physalus patachonica*, was described from the mid-latitudes of the SH [15], but to date, this has not been supported by genetic evidence and is not formally recognised [14,15,16] (Committee on Taxonomy 2022). The three recognised subspecies show a strong divergence between the hemispheres [17]. However, mitochondrial clades are not reciprocally monophyletic between the hemispheres, with genealogies indicative of multiple historical introgressions from the SH into the Northern Hemisphere (NH) and a strong genetic differentiation between the hemispheres and between ocean basins within the NH [16,17,18]. Further, genetic information in the SH is mostly limited to *mtDNA* control region sequences and strongly biased towards Chile, the Antarctic Peninsula and the south of Africa [16,17,18]. However, due to the uneven geographic distribution of genetic samples that are currently available in the SH [16,17,18], it has not yet been possible to fully resolve the SHFW population structure.

For some baleen whale species, acoustic evidence can be used as an alternate method to identify the population structure [19,20,21]. Acoustic data from the western Antarctic Peninsula is indicative of a single, stable fin whale population with limited evidence of acoustic vagrants [22]. Common acoustic 20 Hz pulses (15–30 Hz) are relatively uniform in fin whales for the regions where recordings are available. However, some differences have been observed with the frequency of call overtones differing between the Antarctic Peninsula and Eastern Antarctica, with an increased variation in the groups of individuals east of 115° (99 Hz, 94 Hz and 82 Hz overtones recorded in this region relative to the consistent 84 Hz in the WAP) [22]. To date, circumpolar genetic and acoustic data are unevenly distributed [22]. Therefore, neither approach has been able to fully resolve the fin whale population structure, particularly in the SH.

Some of the lack of resolution in fin whale population genetic studies is associated with the genetic material used for analysis. Given the strong philopatry of female baleen whales to breeding grounds [23], short fragments of the mitochondrial DNA control region (*mtDNA CR*) have been the standard locus used to resolve the population structure in whale species (e.g., [23,24,25]). While historically more accessible, short gene fragments have a lower statistical power to resolve the population structure, especially in species with low *mtDNA CR* diversity (e.g., [26,27]). In recent years, technical advancements in next-generation sequencing have increased the speed and affordability of sequencing multiple samples at the genomic scale, thereby enhancing the power to resolve the population structure [28,29]. Consequently, whole mitogenomes have been increasingly used to identify demographic independence and resolve the population structure in marine megafauna (e.g., green turtles, *Chelonia mydas* [30]; sawfish, *Pristis pristis* [31]; short-finned pilot whales, *Globicephala macrorhynchus* [32]; silky sharks, *Carcharhinus falciformis* [33]). Here, we used whole mitogenomes to better resolve the fin whale population structure in the SH.

Despite extreme exploitation of SHFWs, they have retained a high mitochondrial diversity (estimated using the *mtDNA CR*) in the geographical regions that have been sampled to date (h > 0.99, pi > 0.85%) [16]. This is potentially because the period of the heaviest exploitation, between 1910 and 1970 [11,34], was less than three generation lengths for the species (generation time, 25.9 years) [35] and may have resulted in a population decrease that was not sufficiently long to cause a genetic bottleneck often observed in whale populations as a significant loss of *mtDNA* haplotype lineages (e.g., [36,37,38]). However, recent research has suggested that spatial shifts in connectivity and haplotype diversity may be a frequently overlooked component of biodiversity loss, as the majority of studies focus only on genetic diversity [39]. However, without spatially distributed samples dating back to before the bottleneck and with limited contemporary samples, it is difficult to fully characterise the extent of the bottleneck and its impact on the fin whale population structure.

Large numbers of whalebones were discarded at whaling sites during the 20th century, and some have been curated by museum and historical collections across the globe [40,41]. These samples represent an important historical resource which can be used to understand more about the pre-exploitation populations of whales [40]. Indeed, historic specimens have been used previously to identify changes in genetic diversity, effective population size and population structure of other baleen whale species [42,43,44]. A recent investigation of the *mtDNA CR* genetic diversity using samples from the early whaling period confirms the above hypothesis regarding the strength of the bottleneck due to industrial whaling, showing that historical fin whale diversity was very similar to contemporary diversity [45]

While broad patterns of genetic diversity may be similar between the past and the present, the genetic structure may have changed even over a century of exploitation. For example, a genetic drift might have acted to transiently increase the genetic differentiation of the neighbouring populations temporarily isolated from each other as they were brought down to a low population size [46,47,48]. Additionally, whales may have increased long-distance mobility in response to the lower population size, for example, looking further afield to find mates or moving into areas of higher resource quality during periods of reduced competition, thereby increasing the gene flow over the whaling period. Haplotype distribution and representation may also have changed due to uneven patterns of exploitation [39].

The use of historical samples can therefore provide an invaluable context to better understand the long-term baseline SHFW population structure for comparison with the contemporary patterns. In this study, we used the historical mitogenomes obtained from fin whales hunted at five sites across the western South Atlantic and eastern South Pacific to (i) put the understudied western South Atlantic fin whale population into a global context by comparison with other oceans and (ii) investigate patterns of the historic fin whale gene flow between the western South Atlantic, eastern South Atlantic [17] and eastern South Pacific [16]. These data provide a more nuanced understanding of the genomic population structure of fin whales in the SH, which can be used to better characterise the impacts of industrial whaling and inform future management decisions.

## 2. Materials and Methods

### 2.1. Sample Collection and DNA Extraction

DNA was extracted from historical whalebones previously identified as fin whale (*n* = 50) from the western South Atlantic and eastern South Pacific (*n* = 48; Figure 1) and the North Atlantic (*n* = 2; UK and Ireland). Whole mitogenome (*mito*) and mitochondrial control region (*mtDNA CR*) sequences were generated using next-generation sequencing and polymerase chain reaction (PCR) amplification of a 365 bp fragment of the *mtDNA CR*, respectively. The sample metadata are shown in Appendix A. DNA extractions were carried out in a dedicated ancient DNA laboratory at the University of Cambridge Department of Archaeology following a modified version of the extraction protocol from [49]. Whale samples had not been previously processed at this laboratory (see [50] for further details on sample collection, DNA extraction, and species identification). Briefly, 120–220 mg bone powder (average, 150 mg) or 100–200 mg baleen (chunk from the long edge) per sample were digested at 37 °C for 48–72 h (until the remaining bone powder was not visible to the human eye) or 72 h (baleen only) after a 15 min pre-digestion under the same conditions to remove potential outer sample contamination. High Pure Large Volume Assembly 50 mL (Roche) spin columns were used rather than Zymo Spin columns. After the digestion, DNA was eluted in a TAE buffer (2 × 50 µL through a spin column, final volume, 100 µL) with an incubation time of 5 min at room temperature prior to the final elution. All the other steps followed [49].

### 2.2. mtDNA Control Region Sequencing

Aliquots of DNA extracts (30 μL) from all the 50 samples were transported to the British Antarctic Survey’s (Cambridge, UK) specialised DNA laboratory (Angel 3) for PCR amplification of *mtDNA CR*; no contemporary fin whale samples had previously been processed at this laboratory. The DNA extracts were amplified over varied lengths (850, 500, 400 base pairs (bp)) of the mitochondrial control region using forward primer Tprowhale (5′-TCACCCAAAGCTGRARTTCTA-3′) [51] and reverse primers Dlp8G (5′-GGAGTACTATGTCCTGTAACCA-3′) [52], Dlp5 (5′-CCATCGWGATGTCTTATTTAAGRGGAA-3′) [51] or Dlp4 (5’-GCGGGWTRYTGRTTTCACG-3′) [53], respectively. Amplification was initially tried for the longest base pair length (850 bp) and subsequently for shorter lengths if the longer amplification was unsuccessful. Primer sequences and PCR conditions are shown in Appendix A. All PCRs contained negative controls. PCR amplicons were bidirectionally sequenced on a 3739xl DNA Analyzer at LGC Genomics (Berlin, Germany). Electropherograms were edited by trimming primers and low-quality bases (Phred score < 20), and then the assembled and consensus sequences were aligned using Geneious Prime v2022.2.2 (https://www.geneious.com, accessed on 1 September 2022) using the Geneious alignment algorithm. The *mtDNA CR* alignment was checked by eye and all the sequences were trimmed to match the shortest available sequence.

### 2.3. Next-Generation Sequencing

Of the 50 fin whale extracts, 34 were included for next-generation sequencing (Appendix A). Double-indexed double-stranded DNA libraries were built following [54] using 25 μL DNA extract pre-treated with 5 μL USER enzyme (NED). As genomic DNA of historic samples is naturally fragmented, the initial DNA fragmentation step was not performed. Qiagen Min-Elute kits were used for all purification steps. AmpliTaq Gold DNA polymerase (Invitrogen, Thermo Fisher Scientific, Waltham, USA), was used for index PCRs (*n* of cycles = 16). Negative controls were included during DNA extraction and library-build protocols. The libraries were quantified using an Agilent 2100 Bioanalyzer, pooled at equimolar concentrations and sequenced using an Illumina NextSeq 500 at the University of Cambridge Department of Biochemistry (75 bp paired-end reads).

### 2.4. Bioinformatics and Mitogenome Assembly

The next-generation sequencing reads were demultiplexed into sample-specific fastq files by the University of Cambridge Department of Biochemistry. Full mitogenome assemblies were created as follows: (i) reads of length less than 30 bp were removed; (ii) adapter sequences were trimmed and the overlapping reads collapsed using AdapterRemoval v.2 [55]; (iii) reads were mapped to the fin whale mitogenome (NCBI accession NC-001321) and to the blue whale genome (NCBI accession No. GCA_009873245; used to assess the percentage of endogenous content) using BWA-*mem* [56]. PCR duplicates were marked using SAMtools *markdup* and the collapsed and uncollapsed reads were merged using SAMtools *merge* (SAMtools v1.13) [57]. The reads with a quality score > 20 were retained. PMDtools v0.6 *mapdamage* was then used to identify nucleotide misincorporation rates representing DNA damage within the historic samples [58]. As damage patterns were evident (Appendix A), all the historic reads were trimmed by 5 bp from both the 5′ and 3′ ends, which has been shown to improve the mapping success of endogenous aDNA reads (see [59]). The trimmed reads were then remapped to the fin whale reference mitogenome following the above steps.

Mitogenome assemblies were visually checked for accuracy in Geneious Prime v2022.2.2 and the misaligned reads were subsequently removed. Consensus sequences for all the samples with > 1000 assembled reads and a minimum coverage of 3× were retained and the mitogenomes were aligned (bp = 16,427). To avoid deamination patterns overinflating diversity estimates, the SNP sites that could indicate possible deamination (only present in one historic sample and not present in the contemporary mitogenome dataset from [17]) were removed. For all the mitogenomes, the hypervariable region of the control region was cross-checked for consistency using the *mtDNA CR* obtained from PCRs (see Section 2.2). The mitogenomes were trimmed to the shortest available sequence and the nucleotide sites where at least one sample contained an ambiguity were removed, resulting in sequences of 14,392 bp.

### 2.5. SHFW Diversity and Population Structure

Both the *mtDNA CR* and full mitogenome datasets were used to investigate the genetic diversity and population structure of fin whales in the SH. Genetic diversity was measured using haplotype and nucleotide diversity independently for both loci (*mito*, *mtDNA CR*). Population structure was assessed by comparing haplotype frequencies between groups (F_ST_) and analysis of molecular variance (Φ_ST_). Contemporary genetic data from the literature were combined with the data generated in this study. Contemporary mitogenome and *mtDNA CR* sequences were available online at NCBI (eastern South Atlantic and Indo-Pacific from [17]; Antarctic Peninsula from [18]; eastern South Pacific from [16,17]). The sample sizes are summarised in Table 1 and Table 2. The sequences were associated with one of the two time periods (pre- or post-1986 whaling moratorium; herein referred to as historical and contemporary, respectively) and one of the four SH regions: (i) Antarctic latitudes of the South Pacific (eastern)—SP_e_; (ii) lower latitudes of the South Pacific (eastern)—SP_chile_; (iii) western South Atlantic (historic only)—SA_w_; and (iv) South Atlantic (eastern)—SA_e_. The historical sequences were obtained from two localities (SP_e_ and SA_w_; herein referred to as SA_hist_ and SP_hist_). Contemporary samples were available from three localities (SP_e_, SP_chile_, SA_e_). 

Median-joining haplotype networks of the mitogenomic and *mtDNA CR* datasets were produced using PopART [60]. Genetic diversity metrics, including the number of private alleles within groups, were estimated using Arlequin v.3.5.2.2 [61]. Haplotype diversity (hd) and nucleotide diversity (πd) were estimated using the pre-whaling genetic datasets (*mito*: 14,392 bp; *mtDNA CR*: 241 bp) from two localities (SP_e_; SA_wc_; Table 1 and Table 2) and later combined with a single sample from the SA_e_ to measure the pre-whaling/historical diversity across a third of the SH (southeast Pacific, south Atlantic and western Indian Ocean; Figure 1; *mito*: *n* = 25; *mtDNA CR: n* = 47). Genetic diversity in the contemporary SHFW populations was estimated for three localities (SP_e_, SP_chile_, SA_e_) using both datasets (*mito* 14,392 bp (SA_e_ only); *mtDNA CR* 241 bp (SP_e_, SP_chile_, SA_e_); sample sizes are shown in Table 1 and 2). Additionally, all the contemporary SHFW sequences were combined and used to measure “circumpolar” diversity (*mito* = 43; *mtDNA CR* = 102), although we note here that the samples were spatially patchy (see Figure 1). Statistical differences in the genetic diversity (hd and πd) between the regions were assessed following [62].

Pre-whaling genetic differentiation was assessed between the Pacific (SP_hist_) and Atlantic (SA_hist_) localities using both datasets (for the sample sizes, see Table 1 and Table 2). Contemporary genetic differentiation was assessed between the Pacific (SP_e_ and SP_chile_) and Atlantic (SA_e_) localities using *mtDNA CR* only (for the sample sizes, see Table 2). For the *mtDNA CR* datasets, nucleotide substitution models were identified using jModelTest v.2.1.10 and the Bayesian information criterion [63]. As ambiguities were removed from the protein and non-protein coding regions of the mitogenomes resulting in a length of 14,392 bp (see Section 2.4), genetic differentiation was assessed in the mitogenomes without a nucleotide substitution model. All the tests of genetic differentiation were assessed using the F_ST_ and Φ_ST_ statistics in Arlequin v.3.5.2.2 [61] using 10,000 permutations to test significance. For comparisons within the time periods, both the F_ST_ and Φ_ST_ statistics were analysed. For comparisons between the time periods, only the F_ST_ frequency differences were considered as no molecular differentiation would be anticipated between the time intervals (historical: ~early 1900s; contemporary: 1986 onwards). For the population comparisons that pooled samples over the time periods, only Φ_ST_ molecular differentiation was considered since haplotype frequencies would be confounded. Differences in genetic diversity (hd and πd) and genetic differentiation between the time periods (pre/post-moratorium) were assessed using all the available historical (*mito*: 25; *mtDNA CR*: 47) and contemporary sequences (*mito*: 43; *mtDNA CR*: 102; sample distribution shown in Figure 1). Additionally, Tajima’s *D* [64] and Fu’s *F* [65] were used to look for evidence of population size changes across all datasets.

### 2.6. Comparison with Northern Hemisphere Fin Whales

To assess the genetic diversity and genetic differentiation between ocean basins, we combined all (historical and contemporary) SHFW genetic sequences with those previously sequenced from the North Atlantic (NA_oceanic_) and North Pacific (NP_oceanic_) (sample sizes and NCBI accession numbers are summarised in Table 3) and followed the same methods as described for the interregional comparisons of SHFWs in Section 2.5 using only the Φ_ST_ statistic to investigate longer-term restrictions to the gene flow. For the mitogenomes, this resulted in sample sizes of 16, 96 and 68 for the NA_oceanic_, NP_oceanic_ and SH, respectively. For the *mtDNA CR*, samples were also available from the Mediterranean and the Gulf of California, shown previously to be genetically distinct from the wider oceanic North Atlantic and North Pacific populations, respectively [16,18]. This resulted in sample sizes of 355, 108, 346, 521 and 149 across the five regions (NA_oceanic_, NA_mediterranean_, NP_oceanic_, NP_goc_ and SH; Table 3).

### 2.7. Phylogenetic Analysis

To estimate the evolutionary relationship of the historic mitogenomes within the existing global contemporary mitogenome datasets, a Bayesian phylogeny was built using MrBayes as follows. First, all the mitogenomes were aligned with the contemporary mitogenomes available from [17] (NCBI accession Nos. KC572708 to KC582296) and GenBank (NCBI accession No. NC-001321) [66]. The humpback whale (*Megaptera novaeangliae*) mitogenome (NCBI accession No. NC-006927) [67] was used as an interspecific outgroup following [17]. The best evolutionary model was selected using PartitionFinder v2.1.1 [68]. Each gene was separated into codon positions resulting in 43 partitions (see Appendix A) and input into PartitionFinder using a greedy algorithm. A standard (non-clock) phylogenetic tree was constructed using MrBayes v3.2.7 [69] and the substitution models specified by PartitionFinder. MCMC settings were as follows: four chains, 25 million iterations, 10% burn-in, thinning frequency = 1000. The average standard deviation of the split frequencies was < 0.01 and the minimum estimated sample sizes were > 200 across all the parameters. All the parameters were visually inspected for convergence using TRACER v. 1.7.1 [70]. The majority rule was used to summarise the optimal tree (sumt) visualised using FigTree v.1.4.4 [71].

## 3. Results

### 3.1. Mitogenome and mtDNA CR Sequencing of the Historic Fin Whale Samples

Mitogenome assemblies (*mito*) resulted in near full-length alignments (inclusive of *mtDNA CR*) for 27 samples and full-length alignments (16,407 bp) for 18 samples (see Appendix A). The four resequenced samples had identical *mito* assemblies for both constructions. No sequencing discrepancies were found between the *mito* and *mtDNA CR* sequences within the samples. All the mitogenomes were unique, confirming that they were likely from different individuals (i.e., no replicate samples). After the ambiguities and indels were removed (see Section 2.5), an alignment length of 14,392 bp was used to measure the *mito* genetic diversity and differentiation.

The *mtDNA CR* sequences were successfully amplified for 50 historic fin whale samples over a fragment size of 365 bp. After aligning and trimming all the available *mtDNA CR* sequences from this study and the literature, an alignment length of 241 bp was used to measure the *mtDNA CR* genetic diversity and differentiation. This included 32 sequences from SA_hist_, one from SA_e_, 14 from SP_e_ and three from the NA (Appendix A). JModelTest identified Tamura–Nei with a gamma correction of 0.4 as the most appropriate substitution model for both *mtDNA CR* data.

### 3.2. SHFW Genetic Diversity and Population Structure

#### 3.2.1. Genetic Diversity

Within the historic SHFW *mito* dataset, all the 11 SP_hist_ and 12 out of the 13 SA_hist_ sequences were unique, representing a haplotype diversity of 1.000 (SD = ±0.04) and 0.987 (±0.04) and a nucleotide diversity of 0.001 (±0.001) and 0.002 (±0.001) between the SH regions, respectively (Table 1). Of the historic SHFW *mtDNA CR* sequences, 13 of the 14 SP_hist_ samples and 20 of the 32 SA_hist_ samples were unique within their regions (Figure 2), representing a haplotype diversity of 0.99 (±0.03) and 0.96 (±0.02) and a nucleotide diversity of 0.015 (±0.009) and 0.016 (±0.009) from the two SH regions, respectively (Table 2). Genetic diversity was nonsignificant between the regions (Table 1 and Table 2). Two historical *mtDNA CR* haplotypes were shared between the South Pacific and South Atlantic (Figure 2). Haplotype and nucleotide diversity of the contemporary SHFW *mito* sequences (SA_e_ only) were 0.99 (±0.008) and 0.002 (±0.001), respectively (Table 1). Of the contemporary SHFW *mtDNA CR* sequences (*n* = 102), 33 out of the 46 SAe_e_ samples had unique haplotypes, representing a haplotype and nucleotide diversity of 0.98 (±0.008) and 0.016 (±0.009), respectively (Table 2). Genetic diversity was similar and not significantly different in the two SP *mtDNA CR* datasets, with haplotype and nucleotide diversities ranging between 0.98–0.99 and 0.015–0.016, respectively (Table 2). *Mito* and *mtDNA CR* diversities did not differ significantly between the time periods (Table 1 and Table 2, *p* > 0.05). Tajima’s D and Fu’s Fs were negative for all the regional and combined Southern Hemisphere historical and contemporary datasets, and significant in many cases, indicative of population expansions (Table 1 and Table 2).

#### 3.2.2. Population Structure

Historic SHFW *mito* data provided no evidence for the pre-whaling population structure between the South Atlantic (SA_hist_) and the South Pacific (SP_e_), with pairwise F_ST_ and analysis of molecular variance (AMOVA) both being nonsignificant (F_ST_ = 0.01, *p* = 0.46; AMOVA Φ_ST_ = 0.04, *p* = 0.06). In contrast, *mtDNA CR* provided some evidence for the historical population structure between the SA_hist_ and the SP_hist_; haplotype frequencies differed significantly between the regions (F_ST_ = 0.02, *p* = 0.02), whilst molecular variance did not differ significantly between the regions (AMOVA Φ_ST_ = 0.015, *p* = 0.2). Two of the 16 polymorphic sites identified in the SP_hist_ were private from the SA_hist_. Two of the unique 35 *mtDNA CR* haplotypes were shared between regions (Figure 2). When comparing the historical sequences over a longer length of *mtDNA CR* (365 bp), both F_ST and_ Φ_ST_ were not significant (Appendix A), providing limited support for historical population structuring between these two regions. The contemporary *mtDNA CR* data showed a similar pattern to the historical data, with some evidence for population structure between the South Atlantic and the South Pacific when considered using all the available *mtDNA CR* sequences (241 bp). Haplotype frequencies differed significantly, but not the molecular variance (Table 3, F_ST_ range = 0.03–0.06, *p* < 0.001; AMOVA Φ_ST_ ≈ 0.00, *p* > 0.4).

Using all the SHFW mitogenomes (historical and contemporary), there was no evidence of population structuring between the South Pacific and the South Atlantic (AMOVA only, −0.005, *p* = 0.8). A similar pattern was observed using all the SHFW *mtDNA CR* sequences (AMOVA only, Φ_ST_ = −0.003, *p* = 0.61). When all the SHFW *mito* or *mtDNA CR* sequences were compared between the time periods (historical vs. contemporary), there was a discrepancy between the datasets. No significant difference in haplotype frequencies was observed for the mitogenomes (*mito* F_ST_ = 0.01, *p* = 0.46), whilst a significant difference in haplotype frequencies was observed for *mtDNA CR* (F_ST_ = 0.02, *p* < 0.001). Both time periods displayed high numbers of unique haplotypes, with only one *mito* haplotype shared between the time periods (Figure 3).

### 3.3. Comparison with Northern Hemisphere Fin Whales

Haplotype diversity was lower in the two NH basins relative to the SH across the *mito* and *mtDNA CR* datasets (Table 1 and Table 2). In contrast, nucleotide diversity was similar between the NP (0.0024 ± 0.0012) and the SH (0.0021 ± 0.0010) and lower in the NA (0.0015 ± 0.0008) using the *mito* sequences (Table 4). Conversely, nucleotide diversity was similar between the NA (0.019 ± 0.010 or 0.012 ± 0.007) and the SH (0.016 ± 0.008) and lower in the NP (0.008 ± 0.0010 or 0.001 ± 0.0010) at the *mtDNA CR* level (Table 4). Significant genetic differentiation was observed between ocean basins (AMOVA only, Table 5), and no mitogenomes were shared between the regions (Figure 4), with few *mtDNA CR* haplotypes shared between SHFWs and the NA and the NP (Appendix A). These findings support the previous research showing that fin whales represent independent breeding populations between ocean basins [14,16].

### 3.4. Mitogenome Phylogenetics

Bayesian phylogenetic analysis had high ESSs, well-mixed posteriors, and showed convergence across chains (Appendix A). Global phylogenetics were similar to the previous studies demonstrating a strong association of clades to ocean basins [17,18], but with some slight discrepancies. Posterior probabilities were high across the entire tree, with a strong support for the clades representative of ocean basins (Figure 5). All except one NP sequence fell within two NP clades, one of which clustered more closely with the SH samples (as observed in [17,18]). A single clade representing all except one sample from the North Atlantic (KC572788, see Figure 5) was identified, none of which clustered within the SH (similar to [17,18]). Within the SH, the samples did not cluster by ocean basin (Appendix A).

## 4. Discussion

This study greatly expanded the available genetic sequence data for fin whales and provides the first historical mitogenomes of fin whales. We contributed an additional 50 samples to the global fin whale dataset, including the first genetic samples from the Falkland Islands in the western South Atlantic, totalling approximately 150 available samples for SHFWs to date. As SHFW samples are financially and logistically challenging to obtain, these samples provide a key contribution to what is known about this species.

Despite the exploitation of over 700,000 fin whales from the SH during the 20th century [11,34], high SHFW genetic diversity appears to have been maintained since the end of the industrial whaling. Exploitation of SHFWs was heaviest over a 60-year period between 1910 and 1970 [12]. As fin whales have an estimated generation time of 25.9 years [35], the whaling period lasted for approximately 2–3 fin whale generations. Consequently, the whaling “bottleneck” may not have been sufficiently long to substantially impact genetic diversity in this long-living species or there may be a time lag of diversity loss which is not yet detectable in the contemporary samples. A similar pattern was seen in a previous study that detected little change in genetic diversity in three species of baleen whale at South Georgia and the Antarctic Peninsula before and after the 20th century whaling [45].

### 4.1. Southern Hemisphere Population Structure

The high number of unique haplotypes across both mitogenome and mtDNA CR datasets resulted in statistically significant differences in haplotype frequencies between high-latitude regions of the South Pacific and the South Atlantic. However, levels of genetic divergence were nonsignificant between the areas indicating considerable shared genetic variation between the regions and thus limited SHFW population structure between ocean basins. Many baleen whale species show a pattern of high mtDNA diversity, likely driven by their complex breeding and migration strategy, which can involve breeding in one location, mixing with animals from other breeding sites at a geographically distinct feeding location and potentially also mating with animals from other breeding sites while on migration to and from breeding grounds [72]. Since breeding site fidelity (and for some species, potential feeding site fidelity) is thought to be inherited multi-generationally through maternal lines for most species [73,74,75], this can set up a complex pattern whereby high diversity is maintained long-term through multiple sets of mtDNA lineages being associated with groups of animals (multiple generations) that preferentially visit particular combinations of breeding and feeding sites. Where exploitation had taken place for many centuries (i.e., grey whales, right whales), lower diversity has been observed in the contemporary populations [42,76]. In contrast, for most species hunted predominantly during the 20th-century whaling period, high diversity remains, perhaps, due to the short period of hunting relative to the long life of the species (e.g., [45]).

This complexity of migratory behaviour also influences inference of the population structure. While geographically distinct breeding grounds have been defined for some species (e.g., humpbacks, right whales), neighbouring breeding grounds tend to show low levels of genetic differentiation, reflective of a regular gene flow between regions (e.g., [77,78]). In this study, F_ST_ differentiation between the sites ranged between 0.03–0.06 within the time periods, which corresponds to roughly < 15 migrants per generation (mN_e_); see [72]. However, if the populations being compared are admixed, i.e., one breeding population is compared to a mixture of that population with a second genetically distinct breeding population, this will show significant F_ST_ differences despite a potentially strong influx of the first population into the second area; two examples within humpback whales can illustrate this: significant F_ST_ differences (~0.02) between Tonga (a breeding site) and the Cook Islands (a contributing migratory route) [79,80]; and significant F_ST_ differences between the whales in Gabon (a breeding site) and west South Africa (an associated feeding site and migratory stopover, F_ST_ = ~0.004–0.01) [81]. Here, most of our Southern Hemisphere comparisons were between Southern Ocean sites and the significant levels of F_ST_ coupled with nonsignificant F_ST_ could represent admixtures of multiple breeding groups which are only subtly divergent from one another. However, this F_ST_ range is also in a space where even small variations in estimates can reflect very different rates of interpopulation gene flow [72]. Therefore, we recommend future work aims to randomly sample a minimum of 50 or more individuals within each area to ensure estimates are well-supported (see [82]). Alternatively, further analysis using nuclear SNP data from whole-genome sequencing would provide a higher-resolution perspective and more capacity to detect subtle restrictions to the SHFW gene flow around the Southern Ocean (e.g., methods in [83,84].

Nevertheless, the circumpolar population structuring of SHFWs that we observed here is weaker than that of other species, including humpback whale, *Megaptera novaeangliae*, and southern right whale, *Eubalaena australis*, which both exhibit strong regional population structuring driven by maternal fidelity to breeding and calving grounds [24,38,85,86]. For example, humpback whales breeding off Colombia and feeding in the Antarctic Peninsula (Southeast Pacific) are significantly differentiated from those breeding off Brazil and feeding off South Georgia (Southwest Atlantic) with both mtDNA and nuclear genotypes [87]. Similarly, southern right whales show significant genetic differentiation (mtDNA and nuclear genotype differences) between calving grounds across the South Atlantic and South Pacific [78], although limited genetic sampling in the southeast Pacific to date means that a Southeast Pacific and Southwest Atlantic comparison cannot yet be made. In contrast, the Antarctic blue whale shows a pattern more similar to SHFWs (as observed in the present study). Antarctic blue whales, *Balaenoptera musculus intermedia*, show limited, but significant, mtDNA F_ST_ population structuring on feeding grounds in the Southern Ocean [88,89]. Limited genetic data are available from lower latitudes, and breeding grounds are poorly known [90]. Nuclear DNA was previously used to resolve the SH population structure of Antarctic blue whales (ABW); however, the available data are overall inconclusive about the level of population structuring. One study based on 142 individuals and 20 microsatellite loci suggests that three breeding populations may exist and that Southern Ocean latitudes may in fact consist of mixed assemblages of breeding populations on Antarctic feeding grounds [89]. In contrast, a second study that genotyped 163 individuals showed no evidence for multiple breeding populations, albeit based on six microsatellite loci [88]. Photo identification matches, satellite tracking and discovery mark recoveries suggest ABW can travel substantial longitudinal distances at high latitudes [10,91,92], a mechanism that could facilitate mixing of breeding populations on shared Southern Ocean feeding grounds (as observed by [89]).

SHFWs tend to undertake seasonal, movements between low- and high-latitude areas [8,93]. It is evident from whaling catches that great numbers of fin whales used to inhabit polar and subpolar waters between spring and autumn [12]. Moreover, relative to other species, fin whales were hunted over a wide latitudinal range of 25° and 70° S (highest densities between 40° and 65° S) in summer, indicative of a broad foraging range during the feeding seasons [12]. However, foraging patterns vary between individuals with satellite tracking [93] and isotopic evidence [94] suggesting mixed feeding strategies. Some individuals display isotopic patterns indicative of foraging over a broad latitudinal range whilst others are restricted to their local foraging areas [94]. Alongside varied latitudinal movements between individuals, discovery marks show that SHFWs can travel substantial longitudinal distances at high latitudes, similarly to ABW [9,10]. If fin whales display regional fidelity to wintering grounds (as could be hypothesised from observations from other species (e.g., humpback/SRWs) and the occurrence of fin whales at low latitudes during winter is observed (Chile, Sepulveda et al., 2018; S. Africa) [95], then longitudinal movements may facilitate mixed breeding stocks on high latitudinal feeding grounds; as observed previously for ABWs [89]. SHFW samples used in our study were predominantly collected from subpolar and polar sites (Figure 1), likely to represent fin whale feeding grounds [7,8]. Thus, it is possible that the historical samples used in this study represent mixed breeding stocks, thereby limiting our ability to detect a subtle population structure. Such mixing might be suggested by the observation that fin whales off Chile have significantly different haplotype frequencies from those off the Antarctic Peninsula (Table 5). Given the assumption that both regions (Chile/Antarctic Peninsula) contain members of the same population, with evidence of seasonal migration occurring between these two regions [93], a difference in haplotype frequencies between regions was an unexpected finding in this study, but a result that could be expected if the Antarctic Peninsula represents a feeding ground assembled of mixtures of individuals from distinct breeding stocks.

Similarly to the genetic data, stock structure contradictions have been observed with acoustic data. SHFW acoustic pulses of 15–30 Hz are common and relatively uniform between various SH sites, potentially indicative of a single, stable fin whale population [22]. However, differences in acoustic call overtones have been recognised between the Antarctic Peninsula and Eastern Antarctica [22], with a unique doublet song only recorded off New Zealand to date [96]. These unique acoustic attributes alongside the differences in haplotype frequencies observed in our study make it tempting to hypothesise that SHFWs are not panmictic and may follow a similar mixed feeding ground stocks pattern to that observed for ABWs. Moreover, similarly to ABW, SHFW breeding grounds are poorly known [97]. Genetic samples are currently spatially biased towards IWC management areas I, II and III in the SH (Figure 1; see [8]). Further acoustic and genetic sampling from the western Pacific and from lower latitudes across the SH alongside further knowledge of migration routes (e.g., from satellite tracking or photo IDs) will help to determine whether SHFWs have fidelity to wintering grounds or represent a single circumpolar population.

In this study, sex-based differences in the mitochondrial haplotype frequencies within ocean basins or patterns of genetic differentiation using biparental data (e.g., nuclear SNPs) were not explored. Little is known about the mating systems of fin whales [98]. However, if females have site fidelity to the breeding grounds and males are wide-ranging, as is observed in other species (e.g., sperm whales, killer whales) [62,99,100], an alternative hypothesis for a lack of observed population structure could be long-range movements of males carrying maternally inherited mitochondrial haplotypes throughout the Southern Ocean, making it challenging to detect a subtle population structure. Future work could include whole-genome sequencing of SHFWs and an assessment of sex-specific differences at the nuclear level to identify whether biparental genetic evidence is concordant with mitochondrial patterns.

### 4.2. Global Population Structure

Our results are in concordance with other genetic studies that show limited gene flow between fin whales across the equator [16,17,18], with one study suggesting fin whales should be segregated into three subspecies to capture the strong genetic differentiation between ocean basins [14]. This is unsurprising given the distribution of fin whales is restricted to subtropical to polar waters in both hemispheres (Figure 1) [97]. Trans-equatorial genetic divergence is a pattern also observed in many other baleen whale species, including blue whales, *B. musculus* [101], humpback whales [85,102], right whales, *Eubalaena* spp. [103,104] and sei whales, *B. borealis* [105,106], suggesting that equatorial waters represent a common barrier to gene flow. This pattern is associated with the seasonal and latitudinal migrations that are asynchronous between the hemispheres (summer at high latitudes), reducing the chance of animals encountering those from the other hemisphere populations at low latitudes. We also identified higher levels of haplotypic diversity in the SH relative to the NH populations (Table 5). This pattern has also been observed for other baleen whale species, including humpback, sei, and right whales [85,103,105,106]. Smaller long-term effective population sizes in the NH oceans relative to the SH across species—which is likely to be linked with a lower overall genetic diversity—have been associated with habitat availability following ice advances during the last glacial maximum, a phenomenon which was particularly pronounced in the North Atlantic [107].

It has been suggested previously that two subspecies of fin whales may exist in the Southern Hemisphere [15]. Observations of a smaller darker subspecies of fin whale, the pygmy fin whale, *B. physalus patachonica*, that occur predominantly at lower latitudes relative to its larger counterpart, *B. physalus physalus,* now known as *Balaneoptera physalus quoyi* [14], have been suggested [15]. Previous research resolving the global fin whale subspecies taxonomy found no evidence for this subspecies [16], and the subspecies was recently removed from the agreed list of fin whale subspecies (Committee on Taxonomy 2022). Here, we additionally found no genetic support for a second subspecies (i.e., a distinct lineage) within the SH clades despite sampling within the expected range of *B. p. patachonica* (up to 55° S), thus providing no further evidence to support the existence of *B. physalus patachonica.*

### 4.3. Fin Whale Recovery and Ecosystem Change

SHFWs are wide-ranging generalist predators that feed on a variety of zooplankton and have varied foraging strategies ranging across multiple latitudes [8,94,95,108,109,110]. The substantial longitudinal movements of SHFWs may facilitate increased dispersal relative to other whale species which display stronger site fidelity to breeding and feeding grounds (e.g., humpback whales [111,112]). By displaying a long-ranging dispersal of feeding grounds alongside engulfment foraging strategies [97,113], fin whales have the capacity to impact a wide variety of marine habitats and prey groups relative to smaller, less nomadic and more specialised predators (e.g., penguins, pinnipeds [114,115]). Fin whale recovery since the whaling is now evident with observations of fin whales increasing in the high latitudes of the western South Atlantic and eastern South Pacific in summer. High densities have been reported particularly in the southwest of the Scotia Arc, around the South Orkney Islands, and the Elephant Island near the Antarctic Peninsula, a location where they were also historically abundant based on catch records [7,12,110,116]. Fin whales, among other baleen whales, play a key role in nutrient cycling, through defecation, that can subsequently result in positive feedback loops and increased primary productivity [1]. Therefore, fin whale recovery may positively impact the Southern Ocean (SO) ecosystems.

The SO whale feeding grounds are already undergoing rapid environmental change due to the climate crisis [117,118,119]. We showed here that SHFWs have retained high levels of genetic diversity since the onset of industrial whaling. This, alongside their broad ranging foraging patterns, may give them the adaptive capacity required to respond to the climate crisis relative to other more specialised species (e.g., Antarctic blue whales that specialise on Antarctic krill over a narrower latitudinal range [120]). Continued monitoring of the SHFW genomic diversity alongside an improved understanding of the diet, feeding grounds (summer and winter) and migratory routes is essential in order to understand the impact of fin whale recovery on Antarctic ecosystems and infer the vulnerability of SHFWs to climate change.

## 5. Conclusions

Southern Hemisphere fin whales are key predators in the Antarctic and Subantarctic ecosystems, and an understanding of their genetic identity, migratory behaviour and recovery potential is essential to inform ecosystem management in these regions. This study provides the most comprehensive genetic dataset of SHFWs to date. Genetic analysis supports previous findings that Southern Hemisphere fin whales are genetically distinct from the Northern Hemisphere populations. By using historic and contemporary datasets, our results suggest SHFWs may represent a single panmictic population at the circumpolar scale. However, samples from the Indo–Pacific are currently few, and therefore, additional genetic samples from this region alongside genome-wide assessments of genetic diversity are required to fully resolve the population structure.

## Figures and Tables

**Figure 1 genes-14-01038-f001:**
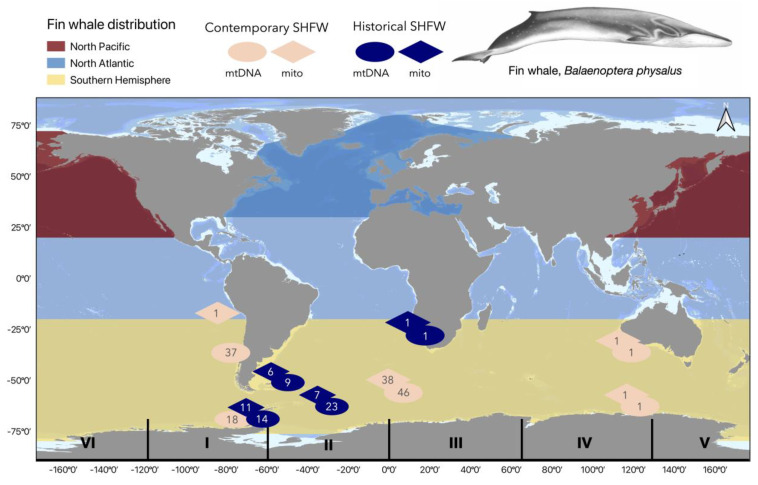
Distribution of genetic samples from Southern Hemisphere fin whales. Historical fin whale samples likely dating to the early 1900s are shown in dark blue. Contemporary fin whale samples (samples collected after the whaling moratorium in 1986) are shown in pink. The number of mitogenome and *mtDNA CR* sequences from a given area are shown within the diamond and circle shapes, respectively. Differences in the described distribution of fin whales within each ocean basin are shown in blue (North Atlantic), red (North Pacific) and yellow (Southern Hemisphere); downloaded from www.iucnredlist.org. Map produced in QGIS 3.26 (www.qgis.org).

**Figure 2 genes-14-01038-f002:**
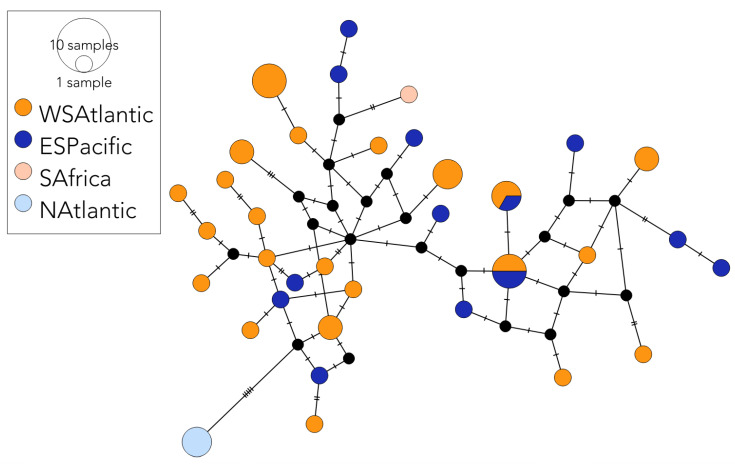
Median-joining haplotype network of the historical SHFW *mtDNA CR* sequences produced in this study. Each circle represents a unique haplotype. The number of nucleotide substitutions between haplotypes are shown by the number of slashes shown on branches connecting circles. Black circles represent unsampled median vectors (see [60]). Sampled haplotypes are coloured by region: western South Atlantic (yellow); eastern South Pacific (dark blue); South Africa (pale pink); North Atlantic (light blue). Sizes of the haplotype nodes are proportional to the total number of samples with that haplotype. Two haplotypes were shared between the regions. Perpendicular lines represent the number of nucleotide substitutions between the haplotypes.

**Figure 3 genes-14-01038-f003:**
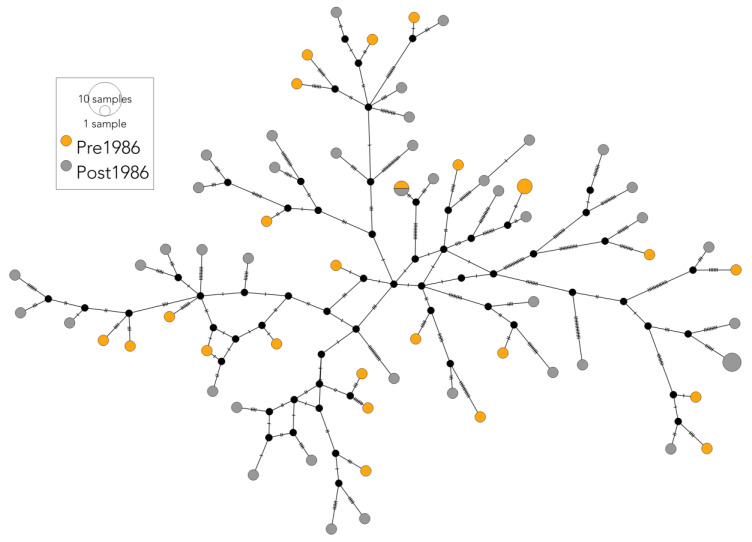
Median-joining haplotype network of the SHFW mitogenome sequences. Haplotypes are coloured by time period: historical (yellow—pre-1986); contemporary (grey—post-1986). Sizes of the haplotype nodes are proportional to the total number of samples with that haplotype. Most haplotypes were unique to a single sampled individual. One haplotype was shared between the time periods. Perpendicular lines represent the number of nucleotide substitutions between the haplotypes.

**Figure 4 genes-14-01038-f004:**
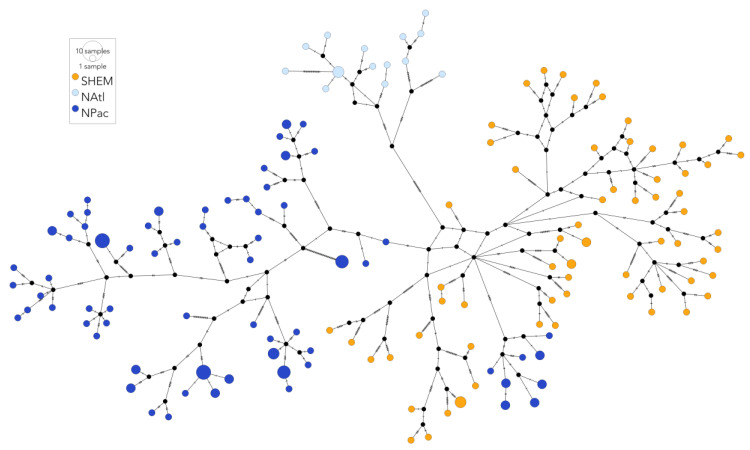
Median-joining haplotype network of the fin whale mitogenome sequences. Haplotypes are coloured by ocean basin: Southern Hemisphere (yellow—SHEM); North Atlantic (pale blue—NAtl); North Pacific (dark blue—NPac). Sizes of the haplotype nodes are proportional to the total number of samples with that haplotype. No haplotypes were shared between ocean basins. Perpendicular lines represent the number of nucleotide substitutions between the haplotypes.

**Figure 5 genes-14-01038-f005:**
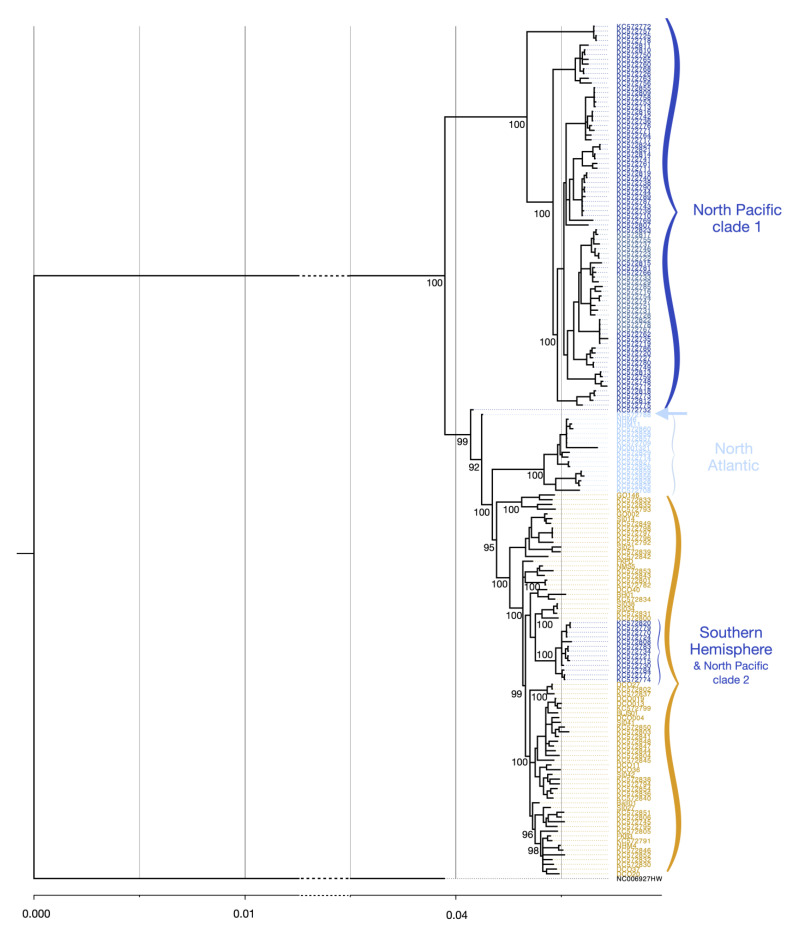
Bayesian genealogy estimated using the fin whale mitogenomic sequences showing good support (> 92%) for the oceanic divergence between ocean basins. One North Pacific clade (dark blue) is nested within sequences from the Southern Hemisphere (yellow). The samples from the North Atlantic are shown in light blue. A single North Atlantic sequence (KC572788 from [17]) did not cluster within the rest of the North Atlantic as denoted by the arrow). Genealogy was created using mrBayes. The humpback whale, *Megaptera novaeangliae*, was used as an interspecific outgroup to root the tree (NCBI accession No. NC_006927). See Appendix A for genealogies with collection site information (Appendix A) and all supporting node values (Appendix A), respectively.

**Table 1 genes-14-01038-t001:** Estimates of haplotype, nucleotide diversity, Tajima’s D and Fu’s F calculated using mitogenome sequences of Southern Hemisphere fin whales (SHFWs). The sequences were trimmed to an alignment length of 14,392 bp. No model of evolution. Numbers of the polymorphic sites unique to a given region are shown within curly brackets. Standard deviations of the diversity estimates are shown in parentheses; *p*-values are shown in square brackets ([*p*]). Hd—haplotype diversity. Nd—nucleotide diversity. Td—Tajima’s D. Circumpolar Hd and Nd did not differ significantly between the time periods (*p* > 0.05, Appendix A).

Population [source]	No. of Samples	No. of Unique Haplotypes	Polymorphic Sites	Hd (sd)	Nd (sd)	Td [*p*]	Fu’s F [*p*]
SHFWs, historical							
Eastern South Pacific (SP_hist_) [This study]	11	11	86 {32}	1.0 (0.04)	0.0014 (0.0007)	−1.39 [0.07]	−2.17 [0.08]
Western South Atlantic (SA_hist_) [This study]	13	12	155 {37}	0.987 (0.04)	0.002 (0.0011)	−1.70 [0.04]	−2.09 [0.10]
Circumpolar [This study]	25	24	222 {66}	0.99 (0.01)	0.002 (0.000)	−2.14 [0.00]	−8.69 [0.00]
SHFWs, contemporary							
Eastern South Atlantic (SA_e_) [17]	38	36	329 {185}	0.996 (0.008)	0.002 (0.0010)	−2.24 [0.00]	−16.55 [0.00]
Circumpolar [17]	43	41	362 {206}	0.99 (0.01)	0.002 (0.000)	−2.30 [0.00]	−20.6 [0.00]

**Table 2 genes-14-01038-t002:** Estimates of haplotype, nucleotide diversity, Tajima’s D and Fu’s F calculated using the mitochondrial control region (mtDNA CR) sequences of Southern Hemisphere fin whales (SHFWs). Sequence alignment length—241 bp. Model of evolution: Tamura and Nei (gamma = 0.42). Numbers of the polymorphic sites unique to a given region are shown within curly brackets. Standard deviations of the diversity estimates are shown in parentheses; *p*-values are shown in square brackets ([*p*]). Hd—haplotype diversity. Nd—nucleotide diversity. Td—Tajima’s D. Circumpolar Hd and Nd did not differ significantly between the time periods (*p* > 0.05, Appendix A).

Population [source]	No. of Samples	No. of Unique Haplotypes	Polymorphic Sites	Hd (sd)	Nd (sd)	Td [*p*]	Fu’s F [*p*]
SHFWs, historical							
Eastern South Pacific (SP_hist_) [This study]	14	14	11 {1}	1.0 (0.03)	0.015 (0.009)	−0.21 [0.50]	−13.9 [0.00]
Western South Atlantic (SA_hist_) [This study]	32	32	17 {1}	1.0 (0.008)	0.016 (0.009)	−0.55 [0.92]	−26.0 [0.00]
Circumpolar [This study]	47	32	20 {2}	0.976 (0.011)	0.016 (0.009)	−0.74 [0.26]	−26.1 [0.00]
SHFWs, contemporary							
Eastern South Atlantic (SA_e_) [17]	46	46	25 {4}	1.0 (0.005)	0.016 (0.009)	−1.26 [0.11]	−26.1 [0.00]
Eastern South Pacific (Ant.Pen) (SP_e_) [18]	18	18	17 {2}	1.0 (0.018)	0.016 (0.009)	−1.14 [0.12]	−20.1 [0.00]
Eastern South Pacific (Chile) (SP_chile_) [16,17]	37	37	16 {3}	1.0 (0.006)	0.015 (0.008)	−0.40 [0.38]	−26.1 [0.00]
Circumpolar [16,17,18]	102	54	30 {12}	0.978 (0.006)	0.016 (0.008)	−1.20 [0.09]	−26.2 [0.00]

**Table 3 genes-14-01038-t003:** Pairwise regional comparisons of SHFW haplotype frequencies (Fst—above the diagonal) and genetic differentiation (Φ_ST_—below the diagonal) of the *mtDNA CR* sequences (241 bp); *p*-values using permutation tests are shown in parentheses.

	Eastern South Pacific pre-1986 (SP_hist_)	Western South Atlantic pre-1986 (SA_hist_)	Eastern South Atlantic post-1986 (SA_e_)	Eastern South Pacific (Antarctic Peninsula) post-1986 (SP_e_)	Eastern South Pacific (Chile) post-1986 (SP_chile_)
Eastern South Pacific pre-1986 (SP_hist_)		0.03 (<0.05)	0.01 (<0.05)	0.05 (<0.01)	0.03 (<0.01)
Western South Atlantic pre-1986 (SA_hist_)	0.004 (0.35)		0.03 (<0.001)	0.06 (<0.001)	0.04 (<0.001)
Eastern South Atlantic post-1986 (SA_e_)	−0.016 (0.78)	−0.00 (0.45)		0.05 (<0.001)	0.03 (<0.001)
Eastern South Pacific (Antarctic Peninsula) post-1986 (SP_e_)	−0.008 (0.56)	−0.015 (0.77)	−0.01 (0.55)		0.06 (<0.001)
Eastern South Pacific (Chile) post-1986 (SP_chile_)	−0.015 (0.70)	0.001 (0.25)	−0.00 (0.43)	0.02 (0.17)	

**Table 4 genes-14-01038-t004:** Estimates of haplotype, nucleotide diversity, Tajima’s D and Fu’s F calculated using the mitogenome and *mtDNA CR* sequences of fin whales within ocean basins. The sequence alignment length used in this study was 14,392 bp. The models of evolution used to calculate diversity are shown for each population. Numbers of the polymorphic sites (P. sites) unique to a given region are shown within curly brackets. Standard deviations of the diversity estimates are shown in parentheses; *p*-values are shown in square brackets ([*p*]). Hd—haplotype diversity. Nd—nucleotide diversity. Td—Tajima’s D.

Dataset [source]	No. of Samples	Unique Haps	P. Sites	Hd (sd)	Nd (sd)	Td [p]	Fu’s F [p]
*Mito* (no model of sequence evolution)							
North Atlantic [17], this study	16	14	87 {64}	0.975 (0.004)	0.0015 (0.0008)	−0.76 [0.22]	−1.29 [0.024]
North Pacific [17]	96	67	326 {251}	1.0 (0.002)	0.0024 (0.0012)	−1.57 [0.027]	−12.78 [0.00]
Southern Hemisphere circumpolar [17], this study	68	64	428 {348}	0.998 (0.003)	0.0021 (0.0010)	−2.34 [0.00]	−24.08 [0.00]
*mtDNA CR* (model of sequence evolution: Tamura and Nei—gamma = 0.42)							
NA—without Mediterranean [17,18], this study	355	72	39 {0}	0.903 (0.009)	0.019 (0.010)	−0.88 [0.21]	−25.3 [0.00]
NA—Mediterranean only [17,18]	108	16	19 {0}	0.784 (0.027)	0.012 (0.007)	−0.85 [0.22]	−26.8 [0.00]
NP—without Gulf of California [16,17,18]	346	33	25 {0}	0.804 (0.015)	0.008 (0.005)	−1.45 [0.05]	−27.5 [0.00]
NP—Gulf of California only [16,17,18]	521	8	7 {3}	0.16 (0.02)	0.001 (0.001)	−1.58 [0.01]	−9.04 [0.00]
South Pacific [This study]	70	41	25 {6}	0.975 (0.008)	0.016 (0.008)	−1.07 [0.14]	−26.2 [0.00]
South Atlantic [This study]	79	46	26 {7}	0.977 (0.006)	0.016 (0.009)	−1.00 [0.15]	−26.1 [0.00]
Circumpolar [16,17,18], this study	149	69	32 {4}	0.979 (0.004)	0.016 (0.008)	−1.16 [0.13]	−26.0 [0.00]

**Table 5 genes-14-01038-t005:** Pairwise genetic differentiation oceanic comparisons of the fin whale populations between ocean basins. Φ_ST_ estimates using the mitogenomes (14,392 bp) are shown below the diagonal and those using the *mtDNA CR* sequences (241 bp) are shown above the diagonal; *p*-values using permutation tests are shown in parentheses.

	North Atlantic (NA)	North Pacific (NP)	Southern Hemisphere (SH)
North Atlantic (NA)		0.29 (<0.001)	0.28 (<0.001)
North Pacific (NP)	0.46 (<0.001)		0.53 (<0.001)
Southern Hemisphere (SH)	0.44 (<0.001)	0.27 (<0.001)	

## Data Availability

The mitogenome and mitochondrial control region sequences generated in this study will be available online at NCBI Genbank (NCBI accession Nos. *mtDNA CR*: OQ884422—OQ884471; *mito*: Appendix A).

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
