# Peer review of "Historical Mitogenomic Diversity and Population Structuring of Southern Hemisphere Fin Whales"

_genes, 2023, doi:10.3390/genes14051038_

Round 1

Reviewer 1 Report

I must sincerely congratulate the authors, because in my opinion it is a great work, well set up, scientifically developed and which fully answers the questions posed. It represents a very important document relating to the genetic diversity of fin whale populations worldwide and a contribution to the reading and interpretation of their subspecific systematics also from a historical point of view.

Author Response

Thank you so much for your positive feedback and fir taking the time to review our manuscript. We are very pleased that you found the manuscript an important contribution to the field.  

Reviewer 2 Report

The manuscript is well written, introduction is sound, study goal is stated clearly and  the methods, which are thoroughly outlined, are suitable to achieve it. Main results of the study are clearly and thoroughly outlined. Discussion is well structured and data interpretation is well argumented. The strongest point of this study is the comparison of genetic data from both historical and contemporary samples, which may help unravel the footprint of human exploitation on the species' genetic diversity. In general, I really enjoyed reading this manuscript. I have only a handful of comments that are listed below:

1) Is it possible to increase the size of Tables 1, 2 and 4 to make them more readable? Alternatively, I think that information on base pairs, model of sequence evolution and source may be moved to caption or footnotes, thus saving room for the remaning columns.

2) Can authors increase the size of the phylogenetic tree that is depicted in Figure 5?

Lines 371:372 - "Mito and mtDNA CR diversities did not differ significantly between time periods (Table 1 & 2, p <0.01*)." Can authors explain this statement? The Probability value, which is well below 0.05, seems to indicate a significant difference for both CR and mitogenome between time periods.

Lines 405:407 - "No significant difference in haplotype frequencies was observed for mitogenomes (mito FST = 0.01, p = 0.46), whilst a significant difference in haplotype frequencies was observed for mtDNA CR (FST = 0.02, p = 0.61)." Here it seems that both mitogenomes and CR showed concordant results: no significant difference in haplotype frequencies. Check this satement for any misprints, please.

Author Response

Thank you so much for taking the time to review our manuscript and for your positive feedback. Thank you also for identifying a few typos in the text. We have checked over the results and have corrected the typos in the text for the p-values relating to Tables 1 and 2. We have also provided p-values and results for temporal comparisons in the supplementary information (S7 and S8). Additionally, we have edited Tables 1, 2 & 4, and expanded Figure 5 as recommended to increase readability. This included moving alignment lengths and repetitive models of sequence evolution to table captions, and moving source details in parantheses under population details. Table 5 has been reformatted to portrait to increase readability. Additionally, we have updated Figure 1 to distinguish the number of samples that were collected from the Falkland Islands from those collected at higher latitudes in the South Atlantic, and made a few grammatical corrections throughout the text (please see attached word document with tracked changes).